# Effects of Persulfate Activation with Pyrite and Zero-Valent Iron for Phthalate Acid Ester Degradation

**Muhammad A. Imran** [1], **Yuzhen Tong** [1], **Qing Hu** [2], **Mingzhu Liu** [1,*] and **Honghan Chen** [1]

[1] Beijing Key Laboratory of Water Resources & Environmental Engineering, China University of Geosciences (Beijing), Beijing 100083, China; maimran@cugb.edu.cn (M.A.I.); wreelmz@163.com (Y.T.); chenhh@cugb.edu.cn (H.C.)

[2] School of Environmental Science & Engineering, Southern University of Science & Technology, Shenzhen 518055, China; huq@sustc.edu.cn

\* Correspondence: liumz@cugb.edu.cn; Tel.: +86-10-82322332; Fax: +86-10-82321081

**Abstract:** Phthalic acid esters (PAEs) are often detected in remediated groundwater using appropriate oxidant materials by in situ groundwater treatment. The study compares zero-valent iron–persulfate with a pyrite–persulfate system to degrade three PAEs—di(2-ethylhexyl) phthalate (DEHP), dibutyl phthalate (DBP), and dimethyl phthalate (DMP). Column experiments were conducted, and rapid oxidation occurred in a pyrite–persulfate system due to sulfate radical generation. DMP concentration was found at about 60.0% and 53.0% with zero-valent iron (ZVI) and pyrite activation of persulfate, respectively. DBP concentration was measured as 25.0–17.2% and 23.2–16.0% using ZVI–persulfate and pyrite–persulfate systems, respectively. However, DEHP was not detected. The total organic carbon concentration lagged behind the $\Sigma_3$ PAEs. Persulfate consumption with ZVI activation was half of the consumption with pyrite activation. Both systems showed a steady release of iron ions. Overall, the oxidation–reduction potential was higher with pyrite activation. The surface morphologies of ZVI and pyrite were investigated using scanning electron microscopy (SEM), energy-dispersive spectroscopy (EDS), and XPS. Intensive corrosion occurs on the pyrite surface, whereas the ZVI surface is covered by a netting of iron oxides. The pyrite surface showed more oxidation and less passivation in comparison with ZVI, which results in more availability of $Fe^{2+}$ for persulfate activation. The pyrite–persulfate system is relatively preferred for rapid PAE degradation for contamination.

**Keywords:** PAEs; persulfate; zero-valent iron; pyrite; column experiment

## 1. Introduction

Subsurface groundwater supplies can be contaminated by a variety of organic compounds, including phthalic acid esters (PAEs), that are released by anthropogenic activities [1,2]. PAEs are a group of chemical compounds that are extensively used as plasticizers and additives in commercial and industrial products [3–5]. PAEs can have harmful effects on human health when acute or chronic exposure occurs via ingestion or dermal contact [6–8]. The ubiquitous presence of PAEs in water and soil is studied by many researchers to understand and to develop sustainable strategies [9–11].

Advanced oxidation processes (AOPs) are promising alternatives for in-situ degradation of organic pollutants [12–14]. Persulfate (PS) activation has more advantages because of higher redox potential ($E^0$ = 2.01 V), stability in subsurface environment, and cost-efficiency [15,16]. Activation of PS can be achieved by many methods to generate sulfate radicals ($SO_4^-$), such as heat [17], UV [18], soluble transition metals [19], activated carbon [20], ozone ($O_3$), hydrogen peroxide ($H_2O_2$) [21], and metal oxides [22]. Fenton like reactions based on the generation of $SO_4^-$ and hydroxide radical ($\cdot OH$) using ferrous ions ($Fe^{2+}$) showed excessive potential in degrading organic contaminants [23,24]. Iron species are widely

used in subsurface in-situ chemical activation of PS because of their availability, cost-effectiveness, ease of application, and catalytic properties. Zero-valent iron (ZVI) and pyrite showed efficient degradation on a wide range of organic pollutants among them [15,25]. ZVI and pyrite are promising PS activators and belong to heterogeneous activation methods explained in Equations (1)–(5).

$$Fe^0 + S_2O_8^{2-} \rightarrow Fe^{2+} + SO_4^{2-}, \tag{1}$$

$$2FeS_2 + 15S_2O_8^{2-} + 15H_2O \rightarrow 2Fe^{3+} + 34SO_4^{2-} + 32H^+, \tag{2}$$

$$2FeS_2 + 7O_2 + 2H_2O \rightarrow 2Fe^{2+} + 4SO_4^{2-} + 4H^+, \tag{3}$$

$$FeS_2 + 14Fe^{3+} + 8H_2O \rightarrow 15Fe^{2+} + 2SO_4^{2-} + 16H^+, \tag{4}$$

$$Fe^{2+} + S_2O_8^{2-} \rightarrow Fe^{3+} + SO_4^- + SO_4^{2-}. \tag{5}$$

Few studies are available on PAEs removal using ZVI-PS [17,26–28]. However, to the best of our knowledge, there is no literature available on the degradation of different-sized carbon chain PAEs using pyrite-PS. In this study, applications of persulfate activation using ZVI and pyrite are compared and categorized for the development of an in-situ subsurface chemical oxidation technology, e.g., permeable reactive barrier (PRB). However, another study concluded that using iron sulfide (FeS) as PRB material has an advantage over ZVI in terms of permeability losses [29]. It is important to study the PS activation ability, generation of $SO_4^-$, PAEs oxidation behavior of ZVI and pyrite during in-situ groundwater treatment. To screen for a more suitable reactive medium for in-situ remediation of PAEs-contaminated groundwater, the study is aimed to compare the degradation of the PAEs with ZVI and pyrite activation of PS. The study also focuses on the consumption behavior of PS, release of iron ions, and solution pH value to analyze the formation of iron corrosion products on the particle surface before and after the reaction.

## 2. Materials and Methods

### 2.1. Materials

Three standard analytical grade PAEs, namely di(2-ethylhexyl) phthalate (DEHP) (99.5%), dimethyl phthalate (DMP) (99.5%), dibutyl phthalate (DBP) (99%), and sodium persulfate (>99.0%) were obtained from Tianjin Fuchen Chemical Reagent Co., Ltd. (Tianjin, China). Analytical grade sodium thiosulfate (>99.0%) was obtained from Shanghai Zhanyun Chemical Co., Ltd. (Shanghai, China). Acetonitrile (high performance liquid chromatography (HPLC) grade) was purchased from MREDA Technology Inc. (Beijing, China). Ultra-pure water by a Millipore Milli-Q system was used to prepare all the solutions. Fine-grained (200 mesh size) ZVI (98.0%) was obtained from Hangda Technology Co. Ltd. (Shenyang, China). Pyrite (95.0%) was obtained from Strem Chemicals, Inc. (Newburyport, MA, USA), then was ground and sieved to 200 mesh size at the China University of Geosciences (Beijing) laboratories.

PAE (DEHP, DBP, and DMP) stock solutions were prepared with a concentration of about 10 mg L$^{-1}$ each. Each stock solution was placed for 1 day in an incubator at a constant temperature of 20 °C and 150 rpm in the absence of light to ensure a complete suspension of PAEs. Quartz white sand of 20–80 mesh size was prepared by rinsing with deionized water. Sieved sand was soaked in 5.0% hydrogen peroxide ($H_2O_2$) for 3 h to remove organic contaminants. Then, the sand was immersed in concentrated hydrochloric acid (HCL) for 24 h to remove the iron oxides. The sand was rinsed with pure water until it reached a neutral pH. The sand was then oven dried at 50 °C and homogenized prior to use.

### 2.2. Experimental Section

A schematic diagram representing the column experimental apparatus is shown in Figure 1. PAEs treatment column tests were performed using a glass column of 25 cm length and 5 cm internal diameter. The columns were filled with a sand and oxidant material (zero-valent iron and pyrite) mixture with a ratio of 1:10 by mass with a length of 21 cm. A homogenized mixture of sand and

oxidant material was filled in 21 increments to ensure the uniform packing. A length of 2 cm above and below the column was filled with sand to avoid channeling. After columns packing, they were purged with nitrogen gas to remove oxygen. Pore volume of reactive column length for ZVI and pyrite was 141 ± 1 mL and 117 ± 1 mL, respectively. Both freshly prepared PAEs and PS solutions were fed by upward flow using peristaltic pumps. Peristaltic pumps were calibrated prior to the run to ensure the equal supply from each solution. The concentration of the feeding solution in the PAEs and PS tank was 10 mg L$^{-1}$ for each PAE and 120 mg L$^{-1}$, respectively. A buffer device was connected at the bottom of the column to feed homogenized solution as shown in Figure 1. The total volume of the buffer device was 0.99 mL. Both feeding solutions diluted each other in the buffer device. After the buffer device, the concentration of the solution of PAEs and PS reduced to 5 mg L$^{-1}$ for each PAE and 60 mg L$^{-1}$, respectively. The concentrations for both solutions at the entrance of the column were taken as initial concentration (C$_0$). Flow rates for ZVI and pyrite soil column were 0.147 mL min$^{-1}$ and 0.122 mL min$^{-1}$. Flow rates were calculated for both columns to set the hydraulic retention time of 8 h because it takes 8 h to feed one pore volume through the reactive column zone. Feeding solutions were replaced twice a week. Nitrogen gas was continuously supplied as a feeding solution throughout the run.

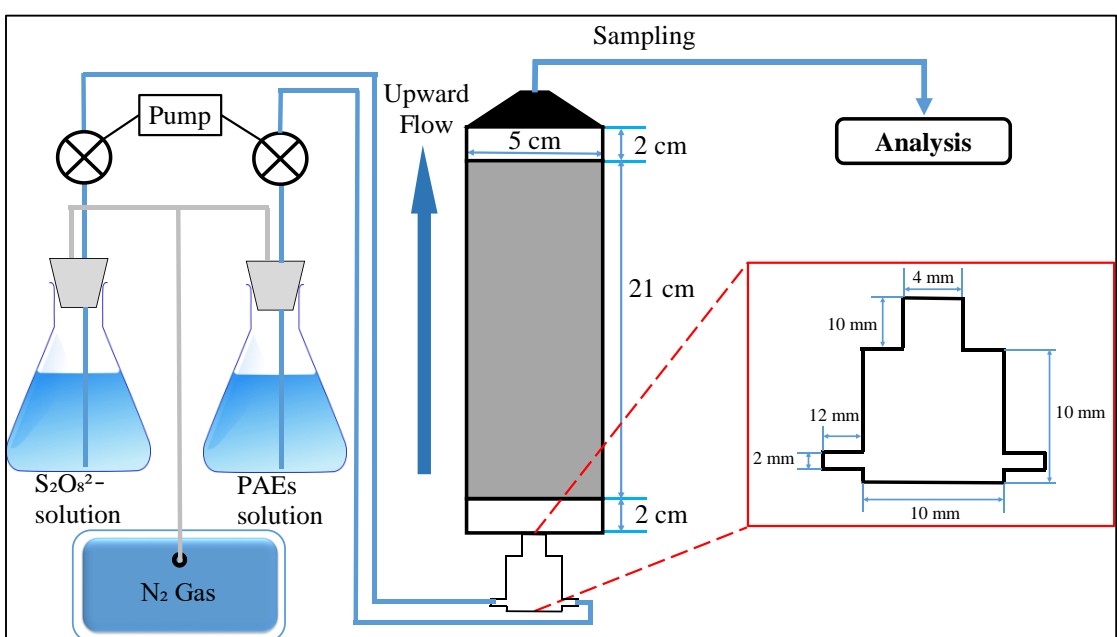

**Figure 1.** Schematic diagram of the experimental apparatus.

*2.3. Analyses and Methods*

High-performance liquid chromatography (HPLC-UV, Agilent 1200 series) equipped with a reverse-phase ZORBAX SB-Aq column (150 mm × 4.6 mm, 5 μm particle size) was used to analyze the PAEs (DMP, DBP, and DEHP). The mobile phase consisted of acetonitrile and water (80:20, v/v). Chromatographic conditions included an injection volume of 30 μL, a flow rate of 1 mL min$^{-1}$, UV detection at 230 nm, and oven temperature at 30 °C. Peaks of DMP, DBP, and DEHP appeared at 2.25 min, 2.95 min, and 6.00 min, respectively, under the operating conditions.

Oxidation-reduction potential (ORP) and pH were measured using an ORP/pH meter. Samples were analyzed immediately after collection for ferrous ions (Fe$^{2+}$), total dissolved iron, persulfate (S$_2$O$_8^{2-}$) ion, sulfate ion (SO$_4^{2-}$), ORP, and pH. The concentration of S$_2$O$_8^{2-}$ was monitored by spectrophotometric determination at 400 nm after adding potassium iodide. The concentration of SO$_4^{2-}$ was measured spectrophotometrically by a SulfaVer4 method (equivalent to United States Environmental Protection Agency (US EPA) turbidimetric method 375.4) using a DR-6000 spectrophotometer. The concentration of ferrous ions (Fe$^{2+}$) was analyzed using a DR-6000

spectrophotometer with 1,10-phenanthroline at a wavelength of 510 nm. Total dissolved iron was determined with hydroxylamine hydrochloride. Total organic carbon (TOC) was determined using a TOC analyzer (TOC-L Shimadzu, Kyoto, Japan). PAEs and TOC samples were filtered through a 0.45 μm nylon membrane filter. Samples were passed through a 0.45 μm polytetrafluoroethylene (PTFE) membrane for ions analysis. PAEs samples were analyzed in duplicate. Sodium thiosulfate was used to quench the $S_2O_8^{2-}$.

The surface morphologies, photographs, and elemental composition of ZVI and pyrite were obtained via scanning electron microscopy (SEM) coupled with energy-dispersive spectroscopy (EDS) by Oxford Instruments Ltd. (Abingdon, England). X-ray photoelectron spectroscopy (XPS) was performed using a Thermo ESCALAB 250Xi, equipped with a monochromatized Al Kα X-ray source to study the surface chemical characteristics. It was operated at a power of 150 W, with a constant pass energy of 20 eV. The peaks were analyzed by using a Shirley background subtraction procedure.

## 3. Results and Discussion

### 3.1. PAEs Degradation

Degradation behaviors of three PAEs (DEHP, DBP, and DMP) by PS activated with ZVI and pyrite are shown in Figure 2a. The pH value of the initial PAEs solution ranged from 6.0–6.1. The decomposition of PAEs occurred in the order DMP < DBP < DEHP with both ZVI and pyrite activating materials. According to our previous findings [27], the oxidation rate of PAEs increased from smaller to longer carbon chain size PAEs. DMP showed the least degradation among these three PAEs. The remaining concentration of DMP was observed at 92.5% and 79.1% on the first day of sampling using the ZVI and pyrite column, respectively. Concentration in the effluent linearly decreased up to day 12 (equal to 36 pv). DMP concentration of 64.8% and 47.9% with ZVI-PS system and Pyrite-PS systems, respectively, remained at 12 days. Afterwards, the concentration of DMP in the effluent became more stable for both systems. The remaining DMP concentration was observed about 60.0% with ZVI and 53.0% with a pyrite activation system at the end of the experiment (day 60). A slight increase in DMP concentration was observed at the end of the experimental time, because of iron oxides formed on the surfaces of the activating materials, which reduced their ability to generate $SO_4^{-}$. A study identified DMP as an intermediate product of DBP [30], which explains the unstable concentration of DMP in the effluent. The remaining DBP concentration ranged from 25.0% to 17.2% with the ZVI-PS system, whereas 23.2–16.0% of the influent DBP concentration remained by a combination of pyrite and PS. Unlike DMP, the concentration of DBP was almost stable throughout the run. The oxidation of DBP established a quick equilibrium with the start of the experiment. In this experiment, most of the PAEs are expected to degrade during the hydraulic retention time. Therefore, after washing with one pore volume of water, the DEHP decomposed completely and its concentration quickly dropped below the detection limit, most of the DBPs degraded and rapidly established chemical equilibrium, resulting in the curves of DEHP and DBP mostly showing static. Whereas, DMP degradation was kinetically controlled which slowly reduced in the first 22 days. After that, the reaction reached equilibrium and the concentration remained stable. Another possible reason for the high concentration of DMP in the effluent at the start of the experiment can be explained as more DMP production as an intermediate product by the rapid degradation of DBP.

Compared with the ZVI-PS system, DMP showed more oxidation in the pyrite activation system. The difference in DMP concentrations was observed more in the early time samples, whereas DBP showed slightly less degradation than the pyrite experiment. DBP concentration was observed relatively lower with pyrite activation than ZVI in early samples. After 22 days, residual DBP concentration was very similar with both activation systems. Eight hours were required to flush one pore volume through the column in our experiment as described earlier. Our previous study [27], as well as Li et al. [26] reported that DBP initially decomposed at a higher rate to a certain level, and then the decomposition rate substantially slowed down. Therefore, the 8-h reaction period was not enough to completely

remove the DBP, but DEHP was totally oxidized during this hydraulic retention time. DEHP was not detected throughout the experimental run for both systems because of the fast equilibrium reaction in this experiment.

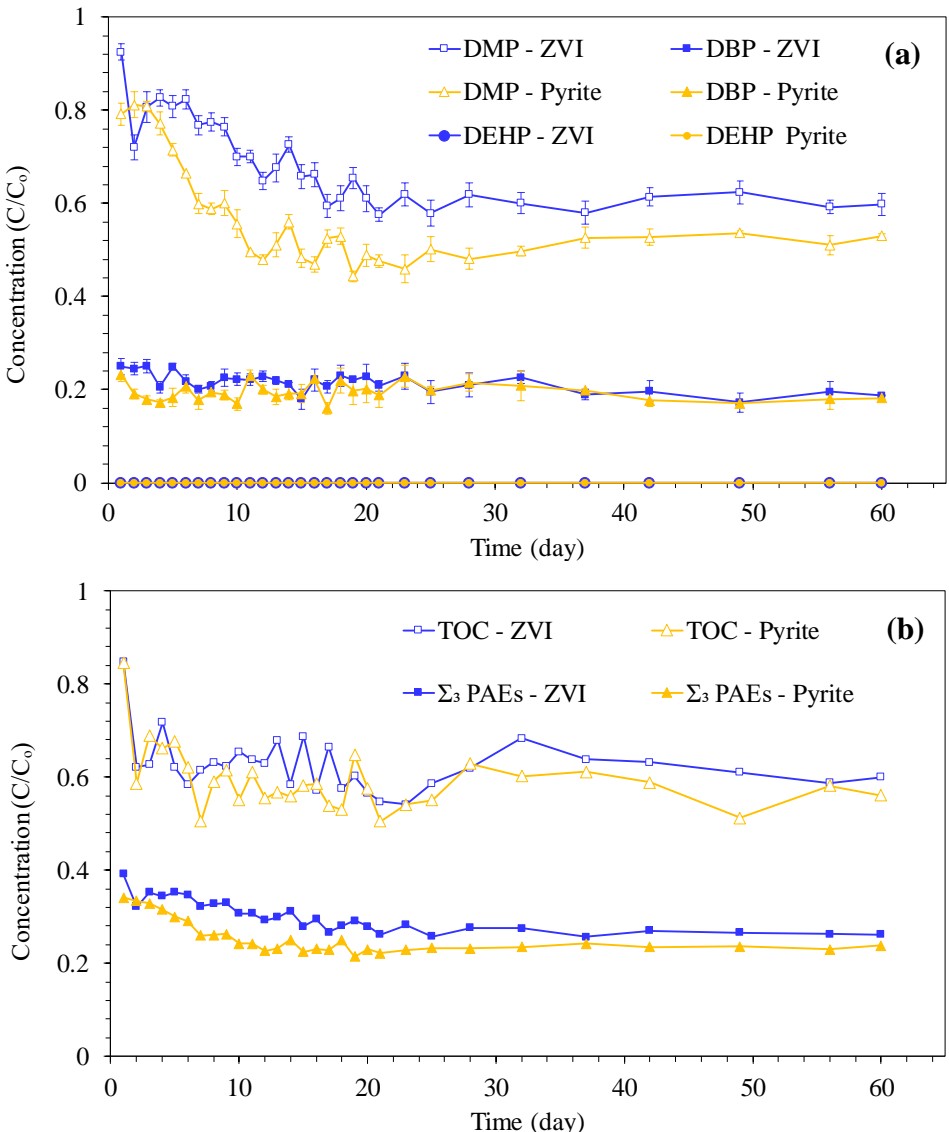

**Figure 2.** (**a**) Residual concentration of individual phthalic acid ester (PAE) in zero-valent iron (ZVI)–persulfate (PS) and pyrite–PS system, (**b**) comparison between residual Σ3 PAEs and total organic carbon (TOC) concentration.

Residual $\Sigma_3$ PAEs (DMP, DBP, and DEHP) and TOC concentration profiles are shown in Figure 2b. The pyrite–persulfate system showed relatively more decomposition of $\Sigma_3$ PAEs throughout the experiment run. According to the results, DEHP completely oxidized and about 20% of DBP of initial concentration remained in the effluent. On the first day of the experiment, the remaining concentration of Σ3 PAEs was 39.1% and 34.1% in ZVI and pyrite experiments, respectively. Lately, the concentration of Σ3 PAEs reduced with the reduction of DMP concentration and stabilized. Difference in the remaining concentration of Σ3 PAEs mainly consisted on the individual DMP concentration. Concentration of $\Sigma_3$ PAEs ranged from 39.1% to 26.5% and 34.1% to 23.7% in ZVI and pyrite experiments, respectively. The $\Sigma_3$ PAEs concentration profile for both systems showed a mild decreasing trend during the run. Similarly, the overall TOC concentration was comparatively lower with the pyrite-activated PS experiment. The concentration of TOC ranged from 71.0% to 54.0% and from 68.0% to 50.0% for ZVI-PS

and pyrite-PS systems, respectively. Comparing the attenuation of TOC with $\Sigma_3$ PAEs, TOC reduction lagged behind the oxidation of PAEs. These findings suggested the organic by-products generated to some degree during the reactions. The fluctuation in the TOC profiles mainly attributed to the PAEs intermediates formations.

Both ZVI-PS and pyrite-PS systems are based on the generation of $SO_4^{\cdot-}$. It is a predominant reactive species that degrades PAEs by attacking the rich electronic site of a molecule. Degradation of the PAEs mainly occurred through hydroxylation of benzene ring and aliphatic chain according to the proposed intermediate pathways in the literature [31]. The degradation of DEHP is initiated by three suggested reactions, including benzene ring hydroxylation, cleavage of the ester bond, and cleavage of side chain [32,33]. Further hydroxylation of hydroxylated DEHP could form mono-(2-ethylhexyl) phthalate (MEHP) or OH-MEHP. The proposed pathways of MEHP intermediates oxidized in the order of MEHP > phthalic acid > benzene > 6-oxohexa-2,4-dienoic acid > lactic acid > carbon dioxide [34]. The reproductive toxicity of MEHP is comparatively less than DEHP [35]. Degradation mechanism and intermediate products of DBP were proposed in another study, which concluded that DBP is predominantly decomposed by an aliphatic chain instead of an aromatic ring [30]. Degradation of DMP mainly occurred in two ways, including hydroxylation of a benzene ring and ester groups attacked by radicals to form carboxylic acid compounds [17,36].

Consumption of ions $S_2O_8^{2-}$ and formation of $SO_4^{2-}$ for both ZVI and pyrite activation are shown in Figure 3. In the ZVI-PS system, the average residual concentration of $S_2O_8^{2-}$ was found at about 28 mg $L^{-1}$ (about 53% of the influent $S_2O_8^{2-}$ concentration), whereas the average $SO_4^{2-}$ concentration of about 26 mg $L^{-1}$ was observed in the effluent. The total concentration $S_2O_8^{2-}$ and $SO_4^{2-}$ was almost 60 mg $L^{-1}$ in the ZVI-PS system. These findings explained that some other products of sulfate formed during the treatment. However, $S_2O_8^{2-}$ were almost completely consumed with the pyrite-PS system. $SO_4^{2-}$ was observed at much higher concentrations for the first 2 days after which the concentration stabilized. The average concentration of $SO_4^{2-}$ was 66 mg $L^{-1}$, which is more than influent $S_2O_8^{2-}$ concentration. The formation of more $SO_4^{2-}$ can be explained by Equation (2), especially in the beginning. Oxidation of sulfur of pyrite mineral into sulfate ions occurred along with the production of $Fe^{3+}$ and hydrogen ($H^+$) ions. There was more production of $Fe^{3+}$ at the beginning as shown in Figure 3, whereas production of more $H^+$ kept the pH value low compared to the ZVI-PS experiment. ZVI-PS showed more efficiency than ZVI-PS in comparing the removal of PAEs and TOC with the consumption of PS. The ZVI-PS system consumed almost half of the PS concentration for the same retention time.

The release of total dissolved iron and $Fe^{2+}$ concentrations is shown in Figure 4. In the ZVI-PS system, the average value of total dissolved iron and $Fe^{2+}$ concentration was 0.58 mg $L^{-1}$ and 0.33 mg $L^{-1}$, respectively. The average value of total dissolved iron and $Fe^{2+}$ for pyrite-PS was 0.59 mg $L^{-1}$ and 0.35 mg $L^{-1}$, respectively. These profiles showed a steady release of iron ions. Just after the supply of PS solution, the concentration of dissolved iron ions was more than 4 mg $L^{-1}$ for the first few pore volumes for both systems. It is attributed to the rapid oxidation of ZVI and pyrite at starting time. Similarly, the pH value in ZVI-PS effluent was about 9.8 at the beginning of the run (Figure 5a), which gradually reduced and became steady in 10 days (30 pv). A similar trend of pH and iron ions was also reported in another study [37]. The pH of the pyrite-PS effluent was relatively lower than the ZVI-PS. However, it is slightly alkaline for both systems. Oxyhydroxides of $Fe^{3+}$ from $Fe^{3+}$ ions at pH > 4.0 were described which showed low efficiency to generate $SO_4^{\cdot-}$ from $S_2O_8^{2-}$ [38]. The corresponding behavior is represented by Equations (6)–(8). The iron oxyhydroxides cover the surface of the activating materials and decrease the supply of $Fe^{2+}$ ions.

$$Fe^{3+} + H_2O \rightarrow FeOH^{2+}, \tag{6}$$

$$Fe^{3+} + 2H_2O \rightarrow Fe(OH)^{2+}, \tag{7}$$

$$2Fe^{3+} + 2H_2O \rightarrow Fe_2(OH)_2^{4+}. \tag{8}$$

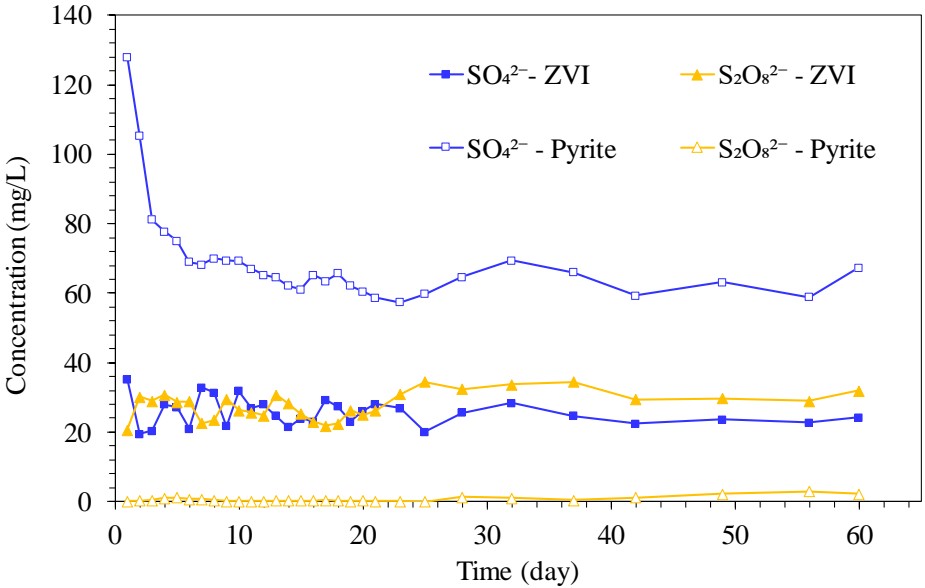

**Figure 3.** Consumption of $S_2O_8^{2-}$ and production of $SO_4^{2-}$ with ZVI and pyrite experiment.

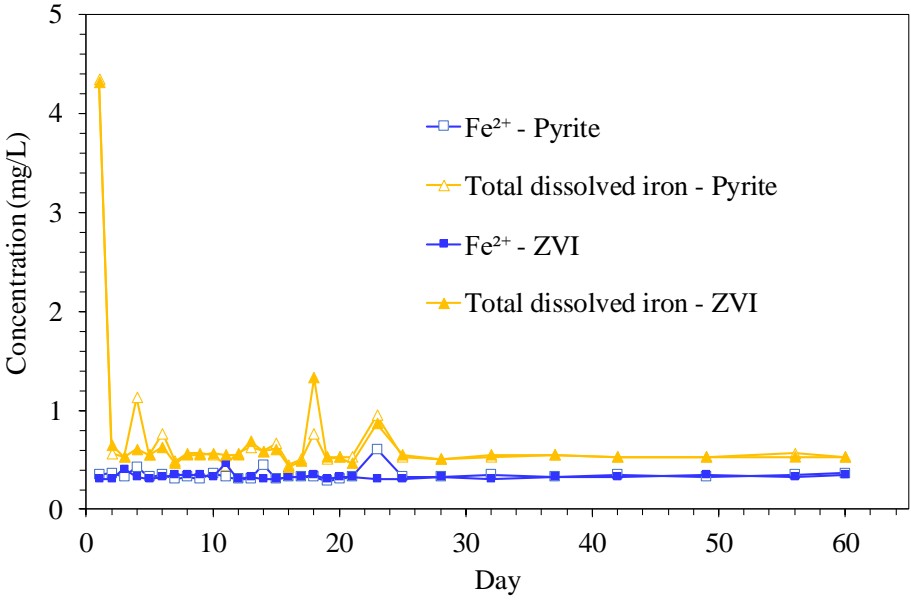

**Figure 4.** Release of total dissolved iron and $Fe^{2+}$ concentration.

Furthermore, $S_2O_8^{2-}$ has the ability to produce more $SO_4^{\cdot-}$ at low pH conditions [39]. The pyrite activation system showed lower pH because of the presence of additional $H^+$. It is because of excessive $S_2O_8^{2-}$ that $SO_4^{\cdot-}$ is generated. In addition, another study reported that $SO_4^{\cdot-}$ and $S_2O_8^{2-}$ can be converted into $HSO_4^-$ and oxygen as explained by Equations (9)–(11) [40]. Both reactions have the ability to occur at the same time as pH is close to 8.0 with pyrite activation.

$$S_2O_8^{2-} + H^+ \rightarrow SO_4^{\cdot-} + HSO_4^-, \tag{9}$$

$$SO_4^{\cdot-} + \cdot O \rightarrow HSO_4^- + \frac{1}{2}O_2, \tag{10}$$

$$S_2O_8^{2-} + H_2O \rightarrow 2HSO_4^- + \frac{1}{2}O_2. \tag{11}$$

The change in the oxidation-reduction potential (ORP) profile in the effluent is shown in Figure 5b. The ORP values are ranging from 133 mV to 239 mV and 146 mV to 252 mV for ZVI-PS and pyrite-PS experiments, respectively. The ORP value is measured generally by the electrode method with a general error of ±10mV. In this experiment, $S_2O_8^{2-}$ was used as an oxidant that generates $SO_4^-$ in the system. The concentration of sulfate radicals exhibits the redox potential of the system [41,42]. The oxidation of sulfate radicals is relatively strong, but its life is very short which results in the fluctuation of ORP value in the whole system. Therefore, the fluctuation of 50 mV was observed in the ORP values. It can be seen from the figure that the ORP value was higher for the pyrite-PS system. A similar trend is observed in $SO_4^-$ production and $\Sigma_3$ PAEs removal while comparing ZVI-PS and pyrite-PS profiles. Difference in ORP values and PAEs removal was more apparent in the start time and gradually reduced toward the end time.

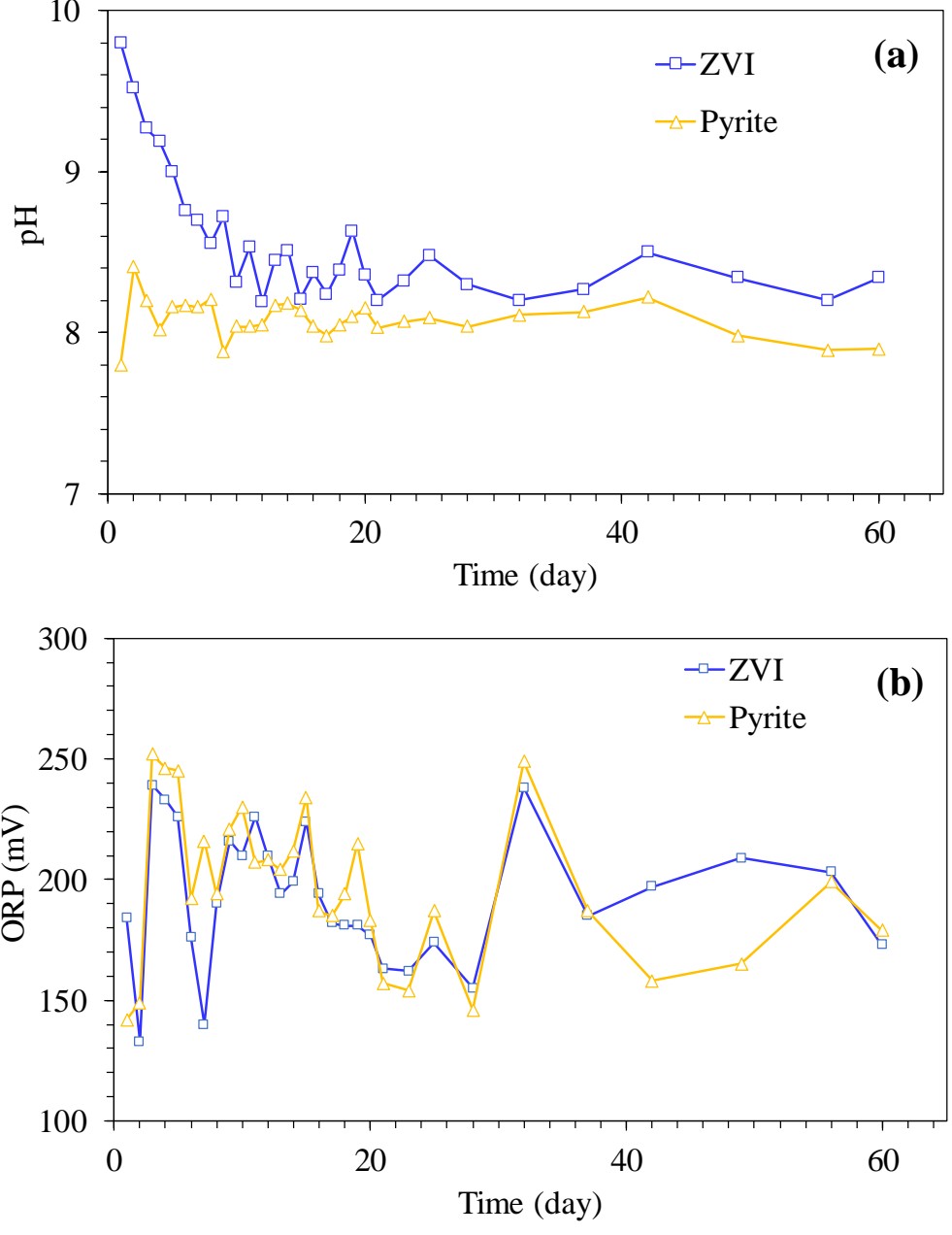

**Figure 5.** (**a**) pH and (**b**) oxidation-reduction potential (ORP) profiles with ZVI and pyrite activation of $S_2O_8^{2-}$.

### 3.2. Surface Characterization of ZVI and Pyrite

The surface morphologies of PS activating materials (ZVI and pyrite) before and after the reaction are shown in Figure 6a–d. SEM photographs reveal that both activating materials were not uniform in shape. Compared with reacted materials, the surfaces of unreacted materials were flatter. After exposure to Σ3 PAEs of 15 mg L$^{-1}$ and PS of 60 mg L$^{-1}$ solutions for 60 days (180 pore volume), their surfaces substantially corroded as shown in Figure 6c,d. Comparing the reacted materials, intensive corrosion occurred on the pyrite surface, whereas the ZVI surface was mainly covered with the net of iron oxides. It is interesting that smaller corroded particles on the surface of pyrite were surrounded by flake-shaped iron oxides that ultimately eroded from the surface due to dissolution and exposed a fresh surface to activate PS. EDS spectra present the elemental compositions of Fe, C, O, and S on the surface of reacted ZVI and pyrite, as shown in Figure 6e,f. The presence of a high percentage of oxygen (O) explains the accumulation of PAEs and their intermediates, as well as surface corrosion products in terms of iron oxides. Iron (Fe) percentage is almost doubled on the pyrite surface compared with ZVI.

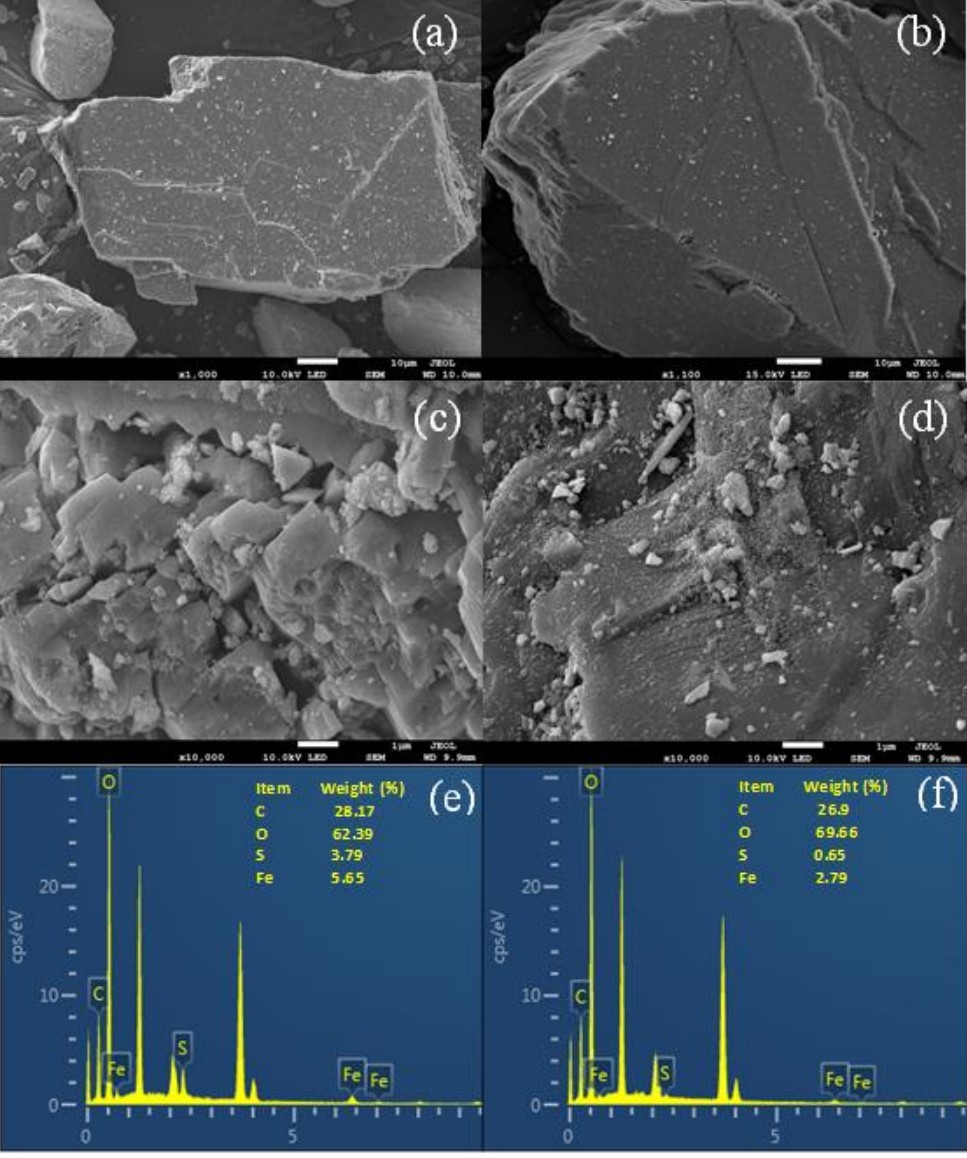

**Figure 6.** Scanning electron microscopy (SEM) images: (**a**) raw pyrite, (**b**) raw ZVI, (**c**) reacted pyrite, (**d**) reacted ZVI and energy-dispersive spectroscopy (EDS) spectra of reacted materials, (**e**) pyrite, and (**f**) reacted ZVI.

Furthermore, the surface characterization of ZVI and pyrite after the experiment was analyzed using X-ray photoelectron spectroscopy (XPS) as shown in Figure 7. The XPS survey spectrum of iron oxide of ZVI-PS and pyrite-PS is shown in Figure 7a. These two spectra reveal the presence of Fe2p, O1s, C1s, and S2p signals. The detailed results for all XPS spectra of Fe2p, O1s, C1s, and S2p were fitted using the Shirley background subtraction method. The Fe2p spectra of an XPS analysis of both pyrite and ZVI are presented in Figure 7b. Only Fe2p$_{2/3}$ was analyzed to evaluate the stoichiometry. The Fe2p$_{2/3}$ spectra of pyrite showed four peaks at 707.1 eV, 710.2 eV, 711.4 eV, and 714.0 eV. A strong peak at about 707.0 eV is representative of the Fe(II)-S species (characteristic of pyrite) [43], while a peak at 710.2 eV is attributed to ferrous oxide, i.e., wusite (FeO)/magnetite (Fe$_3$O$_4$). Another broad peak appearing at binding energy 711.2 eV was related to iron oxides Fe(III)-O, which is a thin precipitate layer on the pyrite particles [21]. The peak at 711.2 eV is recognized as hematite ($\alpha - $Fe$_3$O$_4$) and goethite ($\alpha - $FeOOH), where goethite is dominant in composition [44]. Due to the Fe(III) multiplet structure, a peak was observed at about 714.0 eV. The Fe2p$_{2/3}$ spectra for ZVI showed three peaks. A very short peak at about 707.7 eV consisted of the metallic contribution $\left(Fe^0\right)$. The peak of Fe$^{2+}$ was observed at about 709.9 eV. A peak was observed at binding energy 715.3 eV (between Fe2p$_{2/3}$ and Fe2p$_{1/2}$), which was recognized as Fe$^{2+}$ contribution. The broad peak of Fe$^{3+}$ contribution appeared at the higher binding energy of 711.3 eV. On the basis of results, the peak at 709.9 eV was related to wusite (FeO)/magnetite (Fe$_3$O$_4$), whereas the peak at 711.3 showed the presence of both hematite ($\alpha - $Fe$_3$O$_4$) and goethite ($\alpha - $FeOOH) [45]. Comparing the surface oxidation of activating materials, pyrite oxidation was higher to produce more Fe$^{2+}$. However, passivation of the pyrite surface was less because of a lower pH, which results in a slow FeO production [46], whereas the ZVI surface passivates with more iron oxides and less Fe$^{2+}$ and Fe$^0$ contribution. All binding energies and corresponding full width at half maximum (FWHM) values are presented in Table 1.

**Table 1.** Summary of X-ray photoelectron spectroscopy (XPS) spectrum results.

| Element | Pyrite–Persulfate System | | | ZVI–Persulfate System | | |
|---|---|---|---|---|---|---|
| | Peak Position | FWHM | Assignment | Peak Position | FWHM | Assignment |
| Fe2p | 707.1 | 1.06 | Fe(II)-S | 707.7 | 0.5 | Fe$^0$ |
| | 710.2 | 1.36 | Fe$^{2+}$ | 709.9 | 1.12 | Fe$^{2+}$ |
| | 711.4 | 6.16 | Fe$^{3+}$ | 711.3 | 3.15 | Fe$^{3+}$ |
| O1s | 529.6 | 1.75 | Oxide | 529.9 | 1.72 | Oxide |
| | 531.5 | 1.13 | Hydroxide | 531.5 | 1.31 | Hydroxide |
| | 532.8 | 1.61 | Absorbate | 533.2 | 2.08 | Absorbate |
| C1s | 284.8 | 1.40 | C–C/C–H | 284.8 | 1.34 | C–C/C–H |
| | 286.0 | 2.79 | C–O | 286.2 | 1.31 | C–O |
| | 289.7 | 1.49 | CO$_3^{2-}$ | 289.4 | 1.80 | CO$_3^{2-}$ |
| S2p | 162.4 | 1.18 | S$_2^{2-}$ | – | – | – |
| | 163.6 | 2.00 | S$_2^{2-}$ | – | – | – |
| | 168.5 | 1.15 | SO$_4^{2-}$ | 168.9 | 3.29 | SO$_4^{2-}$ |

"–" not detected.

The detailed spectra of O1s of ZVI-PS and pyrite-PS are depicted in Figure 7c. Peak position for oxides was detected at 529.6 eV of pyrite and 529.9 eV of ZVI. The signal of hydroxides contribution was observed at about 531.5 eV for both reacted materials. The peak of absorbed water oxygen appeared at 532.8 eV and 533.2 eV for pyrite and ZVI, respectively.

C1s showed three peaks for both systems (Figure 7d). The peak appeared at 284.8 for both ZVI and pyrite, attributed to C–C and C–H. Another peak also provides information of the presence of hydrocarbons and its intermediates (C–O) at 286.0 eV and 286.2 eV on the surface of pyrite and ZVI, respectively. It is notable that the peak at around 286.0 eV on the pyrite surface is less intensive than that for ZVI, which is attributed to less iron oxide on the pyrite surface. The peaks for surface carbonates were observed at about 589.7 eV of pyrite and 589.4 eV of ZVI.

The XPS spectra of S2p are presented in Figure 7e. The peak positions of S2p at binding energy 162.4 eV and 163.6 eV attributed to $S_2^{2-}$ under the pyrite-PS system. The S2p component at 168.5 eV of pyrite and 168.9 eV of ZVI represent the binding energy of $SO_4^{2-}$. It indicates formation of sulfate salts on the activating materials. Comparing with pyrite, $SO_4^{2-}$ on the ZVI surface is very small, which means less consumption of PS.

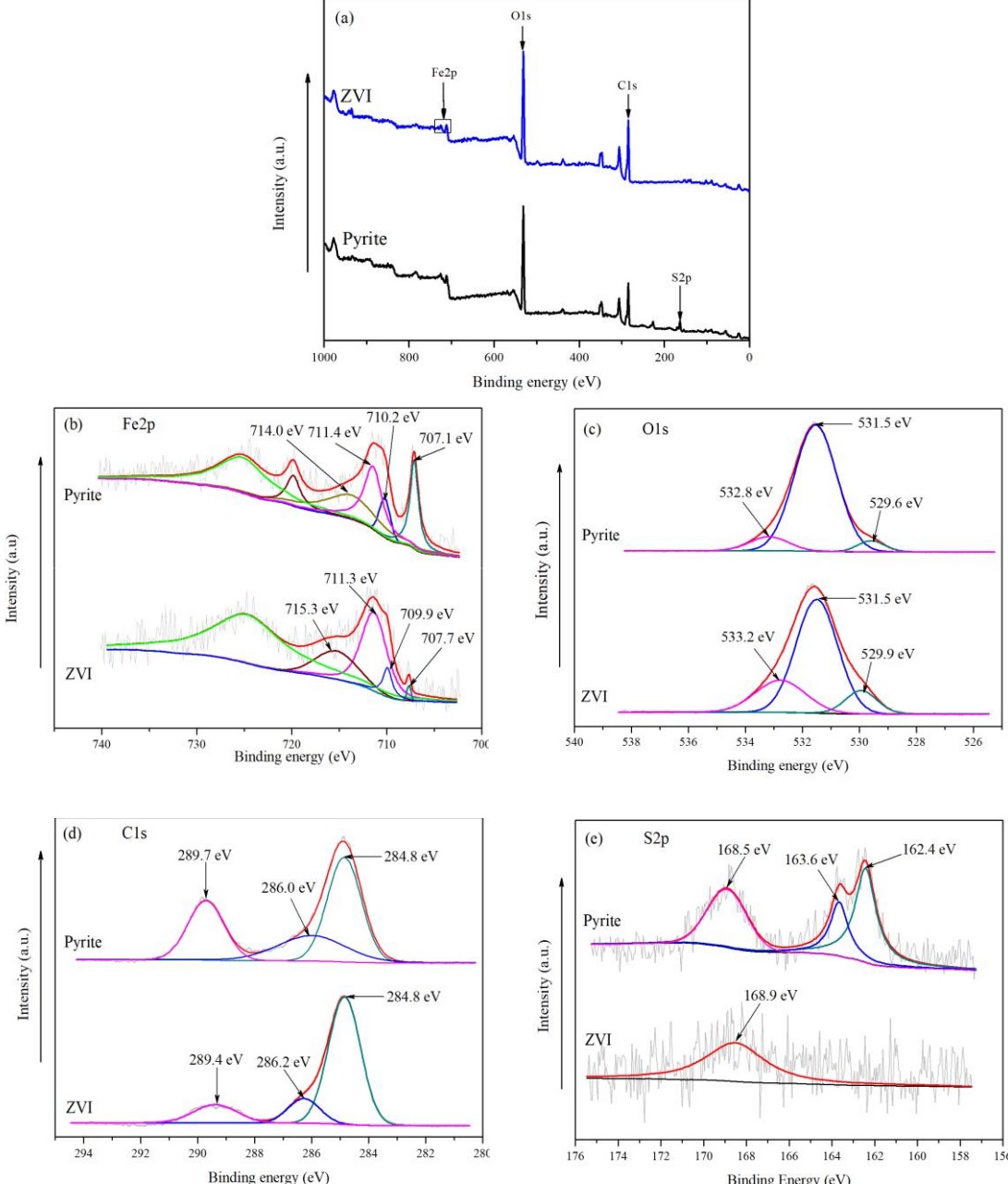

**Figure 7.** XPS spectra of ZVI and pyrite surface layer after the reaction. (**a**) full-range scan of the samples, (**b**) Fe2p spectra, (**c**) O1s spectra, (**d**) C1s spectra, (**e**) S2p spectra.

## 4. Conclusions

The present study compared the degradation of three phthalic acid esters (PAEs), di(2-ethylhexyl) phthalate (DEHP), dibutyl phthalate (DBP), and dimethyl phthalate (DMP), in column experiments using persulfate activation with zero-valent iron–persulfate and pyrite. In addition, persulfate $(S_2O_8^{2-})$ consumption, the release of sulfate ions $(SO_4^{2-})$, total dissolved iron, $Fe^{2+}$, and pH values were

observed. Furthermore, the surface characterization of persulfate activating materials was also studied. The column experiments were conducted on similar conditions of 60 mg L$^{-1}$ persulfate solution, and 5 mg L$^{-1}$ of each PAEs concentration was fed to the columns. The influent flows were fixed to an 8-h retention time according to their pore volume. The results showed more oxidation of PAEs and TOC removal under the pyrite–persulfate system. PAEs degradation was observed in the order DMP < DBP < DEHP. In a hydraulic retention time, the reaction of oxidation of DEHP reached equilibrium instantly, DEHP was completely degraded and the concentration was not detected throughout the experiments under both systems. Moreover, the reaction of DBP degradation, not completely, quickly reached equilibrium but the concentration of the remaining DBP ranged from 25.0% to 17.2% and 23.2% to 16.0% with ZVI and pyrite activation systems, respectively. After the kinetical reaction, the remaining DMP concentration at the end of the experiment was observed about 60.0% and 53.0% with ZVI and pyrite activation, respectively. The concentration of TOC in the effluent was observed from 71.0% to 54.0% and 68.0% to 50.0% under ZVI-PS and pyrite-PS systems, respectively. It was also concluded that the ZVI–persulfate system is more efficient in terms of less persulfate consumption to decompose relatively more PAEs. According to scanning electron microscopy-energy dispersive spectroscopy (SEM-EDS) analysis, intensive corrosion occurred on the surface of pyrite, which provides a fresh surface for persulfate activation, whereas the ZVI surface was covered by a net of iron oxides that reduced the activation ability of ZVI. With XPS analysis, it was observed that more oxidation and less passivation of pyrite occurred compared to ZVI. From this study, it can be concluded that both persulfate activating materials can be efficiently applied for in situ groundwater remediation. The pyrite–persulfate system may be preferred for rapid oxidation of PAEs at an intensively contaminated site because of the nature of pyrite that is easier obtainable and has a longer lifetime.

**Author Contributions:** Conceptualization, M.A.I., M.L. and H.C.; formal analysis, M.A.I. and Y.T.; investigation, M.A.I.; methodology, M.A.I., Y.T., M.L. and H.C.; resources, M.L.; supervision, M.L. and H.C.; validation, Q.H. and M.L.; visualization, M.L. and H.C.; writing—original draft, M.A.I.; writing—review & editing, Q.H. and M.L. All authors have read and agreed to the published version of the manuscript.

**Funding:** This research was funded by China Major Science and Technology Program for Water Pollution Control and Treatment [2018ZX07109-002,2009ZX07424-002], Beijing Natural Science Foundation [8181002].

**Acknowledgments:** This material is based upon work supported by the Beijing Key Laboratory of Water Resources & Environmental Engineering, China University of Geosciences (Beijing). The authors would also like to thank the editor and anonymous reviewers for their valuable comments and suggestions to improve the quality of the paper.

**Conflicts of Interest:** The authors declare that there is no conflict of interest regarding the publication of this paper.

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
