# Peer review of "Effects of Persulfate Activation with Pyrite and Zero-Valent Iron for Phthalate Acid Ester Degradation"

_water, doi:10.3390/w12020354_

Round 1
Reviewer 1 Report
The manuscript entitled „Effects of Persulfate Activation with Pyrite and Zero-Valent Iron for Phthalate Acid Esters Degradation” reports results on the heterogeneous catalytic oxidation of phthalic acid esters by peroxydisulfate. Unfortunately, there are numerous aspects of the experimental work that needs clarification, because in their present state they do not support the conclusions. The primary results of the column experiments are controversial.
1) The primary experimental results are shown in Fig. 2, but why do the DBP curves start at c/c0 = 0.2 (Fig. 2A)? Accordingly, why do the summa PAEs curves start at c/c0 = 0.4 (Fig. 2B)? What happens in the system in the first day of the reaction? The reported curves mostly show a static, unreactive state of the systems, thus, do not convey any information on the oxidation reactions.
2) The oxidizing agent peroxydisulfate is supposed to either remain unreacted or yield 2 equivalents of sulfate as the reduced product. If the corresponding peroxydisulfate and sulfate pairs are summarized by using this stoichiometry in Fig. 3, they do not give back the original 140 mg/L concentration in any time point. This means that mass balance is incomplete. Where is the deficient peroxydisulfate? This deficiency is especially intriguing since the system is already static at these time points (cf. comment No. 1).
3) The ca. 50 mV uncertainty in Fig. 5b suggests that there is some serious matrix effect or other interference in the ORP measurements that make it unreliable.
Reviewer 2 Report
The presence of phthalates in groundwaters is a huge problem due to the potential damages that these pollutants can cause to humans. Hence, effective remediation technologies are necessary for the removal of these hazardous compounds from natural waters.
The present manuscript focused on the study of degradation of three phtalic acid esters (PAEs) by oxidation processes promoted by zero-valent iron/persulfate and pyrite/persulfate, in order to evaluate the feasibility of these systems as Permeable Reactive Barriers.
The topic of the manuscript is suitable for publication on Water.
The experimental part is clearly explained, different techniques were employed and interesting results were obtained and discussed.
The language is clear and correct.
In the light of the above, in my opinion the present manuscript can be accepted after some minor revisions. In the following, some particular comments.
Did the Authors investigated the degradation behavior of PAEs in presence of other compounds, such as natural organic matter (NOM) or salts, in order to evaluate the possible interference on pollutants degradation in real aquatic systems? 95-96. Did the Authors test other concentrations level of PAEs, in addition to 10 mg/l? What was the pH of the initial solution of PAEs? LL 131-132. The Authors write that Figure 2a shows the degradation behaviors of the three PAEs, but only DMP and DBP are reported. It should be specified that DEHP is not shown because it was never detected ( 156-157).Author Response
Please see the attachment.

Reviewer 3 Report
Dear Authors, thanks for your very solid work and the very nice presentation of a very complex topic. Slight improvement of language may be possible but the paper is very well readable. It´s a pity that the results show only small differences between both approaches.
Round 2
Reviewer 1 Report
The authors did a fine job of improving the technical quality of the manuscript entitled "Effects of persulfate activation with pyrite and zero-Valent iron for phthalate acid esters degradation". I accept their responses.
My only remaining concern is, that the only parts of the responses (Response No. 1, 2 and 3 in the Cover Letter) are incorporated into the revised manuscript. I suggest to include the FULL TEXT of the responses in the revised manuscript, as this would further improve its quality and readability.
